# Sodium Phenylbutyrate Ameliorates Inflammatory Response Induced by *Staphylococcus aureus* Lipoteichoic Acid via Suppressing TLR2/NF-κB/NLRP3 Pathways in MAC-T Cells

**DOI:** 10.3390/molecules23123056

**Published:** 2018-11-22

**Authors:** Xin Wang, Mengmeng Zhang, Ning Jiang, Aizhong Zhang

**Affiliations:** College of Animal Science & Veterinary Medicine, Heilongjiang Bayi Agricultural University, Heilongjiang, Daqing 163319, China; byndwx@126.com (X.W.); byndzmm@163.com (M.Z.); jiangng_2008@sohu.com (N.J.)

**Keywords:** sodium phenylbutyrate, lipoteichoic acid, inflammation, TLR2, NF-κB, NLRP3

## Abstract

This study aimed to investigate the anti-inflammatory properties of sodium phenylbutyrate (SPB) against *Staphylococcus aureus* (*S. aureus*) lipoteichoic acid (LTA)-stimulated bovine mammary alveolar (MAC-T) cells. Quantitative PCR was performed to examine the effect of SPB on inflammatory cytokines and host defense peptide (HDP) gene expression. Western blot wanalysis was used to detect the effect of SPB on the TLR2/NF-κB/NLRP3 signaling pathway. The results showed that SPB significantly suppressed the expression of TNF-α, IL-1β, IL-6; meanwhile, the markedly decreased expression of LTA-stimulated TLR2, NLRP3, ASC, caspase-1, and IL-1β, and the inhibited IkBα and p65 phosphorylation were also observed. However, increased TAP and Bac5 expression in LTA-stimulated MAC-T cells was further detected. In summary, these results suggest that SPB ameliorates the inflammatory response induced by *S. aureus* LTA via suppressing the TLR2/NF-κB/NLRP3 signaling pathway, which indicates that SPB may be a potential agent for the treatment of bovine mastitis.

## 1. Introduction

Bovine mastitis, the most prevalent disease of dairy cattle worldwide, has huge effects on the farm economy due to the decline in milk production and quality [1]. *Staphylococcus aureus* (*S. aureus*) is one of the major infectious pathogens responsible for mastitis in dairy cows [2]. It colonizes the mammary gland tissues by internalizing epithelial and endothelial cells. *S. aureus* produces virulence factors such as enterotoxins, toxic shock syndrome toxin, and Panton–Valentine leucocidin, which can stimulate severe host cell inflammatory responses and enhance the expression of various pro-inflammatory mediators in bovine mammary gland epithelial cells [3]. Although the expression of pro-inflammatory cytokines is important for eliminating invasive bacteria, excessive production of these pro-inflammatory mediators often leads to a continuous inflammatory response in bovine mammary glands, possibly resulting in chronic mastitis [4]. Lipoteichoic acid (LTA) is the main component of the cell wall of Gram-positive bacteria [5]. Recent studies reported that LTA plays an essential role in the pathogenesis of *S. aureus* by participating in its adhesion and colonization, and by stimulating the inflammatory response of host cells [6]. Abundant research indicated that *S. aureus* LTA could induce TLR2 activation and subsequently lead to the stimulation of NF-κB signaling and NLRP3 [7,8]. The NLRP3 inflammasome is tightly controlled by a priming step that is dependent on NF-κB and is highly inducible in response to pro-inflammatory stimuli such as *S. aureus* LTA [9]. The NLRP3 inflammasome consists of NLRP3, apoptosis-associated speck-like protein containing CARD (ASC), and pro-caspase-1, which influences the regulation of pro-inflammatory cytokine secretion, particularly the generation of interleukin 1 beta (IL-1β) [10]. These inflammatory cytokines cause damage to mammary tissues [11].

Butyrate is a major short-chain fatty acid (SCFA) produced by bacterial fermentation of undigested dietary fiber in the colon and anterior stomach of ruminants [12]. As an inhibitor of histone deacetylase (HDAC), butyrate not only functions as an energy source, but also plays an anti-inflammatory role [13]. Sodium phenylbutyrate (SPB) is a salt of an aromatic fatty acid, also known as 4-phenylbutyrate (4-PBA) or 4-phenylbutyric acid [14]. Considering that sodium butyrate (SB) is a pungently smelling compound, odorless analogs such as SPB are more appropriate for livestock production. Previous work proposed that SPB promotes the development of a subset of human dendritic cells, which increases the expression level of human cathelicidin and enhances the killing capacity of *S. aureus* [15]. The report also suggested that SB activated bMECs via TLR2/p38, which increased host defense peptide (HDP) expression before/after *S. aureus* invasion, and exerted anti-inflammatory effects during infection [16]. However, the anti-inflammatory effects and molecular mechanisms of SPB on *S. aureus* LTA-stimulated bovine mammary alveolar (MAC-T) cells remain to be elucidated.

The aim of this work was to examine the anti-inflammatory effects of SPB in *S. aureus* LTA-stimulated MAC-T and to investigate potential mechanisms.

## 2. Results

### 2.1. The Effect of SPB on Cell Viability

The cytotoxicity of SPB on cell viability was evaluated by MTT assay after incubating MAC-T for 24 h. As shown in Figure 1, the results showed that SPB (0.5, 2, 4 mM) has no cytotoxic effects on MAC-T. Therefore, in the subsequent studies, the doses of SPB were chosen as 0.5, 2, and 4 mM.

### 2.2. The Effect of SPB on Inflammatory Cytokines and HDP mRNA Expression in LTA-Stimulated MAC-T

To investigate the anti-inflammatory effects of SPB, the levels of pro-inflammatory cytokines were detected by qPCR. As shown in Figure 2, the results showed that LTA significantly upregulated the gene expression of TNF-α, IL-1β, and IL-6 in comparison with untreated cells. SPB slightly increased the mRNA expression of TNF-α, IL-1β, and IL-6. However, SPB suppressed TNF-α, IL-1β, and IL-6 expression in LTA-stimulated MAC-T.

To confirm whether SPB modulates the gene expression of HDP in MAC-T (stimulated with or without *S. aureus* LTA), total RNA isolation and qPCR were performed. Untreated and unstimulated cells showed basal TAP and Bac5 mRNA expression. As shown in Figure 2, MAC-T stimulated with LTA increased TAP and Bac5 mRNA levels in comparison with unstimulated cells. With regards to cells only treated with SPB, TAP and Bac5 mRNA expression was markedly increased. Interestingly, SPB further increased TAP and Bac5 expression in LTA-stimulated MAC-T.

### 2.3. The Effect of SPB on LTA-Induced TLR2 Expression

To further investigate the anti-inflammatory mechanism of SPB, the expression of TLR2 was detected by Western blot analysis. The results showed that LTA treatment significantly upregulated the expression of TLR2. However, pretreatment with SPB significantly inhibited LTA-induced TLR2 expression in a dose-dependent manner (Figure 3).

### 2.4. The Effect of SPB on NF-κB Signaling Pathway Activation

To investigate whether NF-κB is involved in the pathway via which SPB regulates the inflammatory response, we evaluated the phosphorylation status of IkBα and p65. The results showed that LTA treatment significantly activated the NF-κB signaling pathway. Pretreatment with SPB significantly inhibited the phosphorylation of IkBα and p65 in LTA-stimulated MAC-T (Figure 4). These results suggest that SPB inhibited the NF-κB activation induced by LTA in a dose-dependent manner.

### 2.5. The Effect of SPB on NLRP3 Inflammasome Activation

To investigate the anti-inflammatory mechanism of SPB, the effect of SPB on the NLRP3 signaling pathway was measured. As shown in Figure 5, the expression of NLRP3, ASC, caspase-1, and IL-1β was significantly increased in the LTA-treated group. However, treatment with SPB significantly inhibited LTA-induced expression of NLRP3, ASC, caspase-1, and IL-1β in a dose-dependent manner.

## 3. Discussion

Uncontrolled mastitis in dairy cows is usually caused by bacterial infection of the mammary gland and is always accompanied by various complications [17,18]. Bacterial toxins can break through the blood–milk barrier and damage other organs [19]. In addition, bacteria may be a source of food poisoning due to their tendency to produce enterotoxins that remain active after heat treatment [20]. LTA, a bacterial endotoxin embedded in the cytoderm of *S. aureus*, was identified to activate inflammatory responses [21]. A previous study demonstrated that butyrate reduces *S. aureus* internalization into bMECs and regulates HDP gene expression [22]. However, the molecular mechanism of SPB in *S. aureus* LTA-stimulated MAC-T remains unclear. In the present study, MAC-T cells were stimulated by *S. aureus* LTA with or without SPB pretreatment, to investigate the effect of SPB on the inflammatory response.

Cytokines, an essential component of inflammatory mediators, play a crucial role in the process of inflammation. Hong et al. [23] found that SPB decreased the levels of pro-inflammatory TNF-α and IL-1β in serum of rat with acute pancreatitis. Zeng et al. [24] showed that SPB reduced the release of the pro-inflammatory mediators, IL-1β, TNF-α and IL-6, and significantly inhibited lipopolysaccharide (LPS)-activated endoplasmic reticulum (ER) stress in A549 cells. Xu et al. [25] demonstrated that SPB can suppress the expression of TNF-α, IL-1β, and IL-6, and can reduce neutrophil infiltration in mice liver. In the present study, as a kind of nutritional compound, SPB had no obvious inflammatory responses in MAC-T cells; however, pretreatment with SPB significantly decreased the expression of three pro-inflammatory cytokines (TNF-α, IL-1β, and IL-6) in LTA-stimulated MAC-T, which is in agreement with previous studies [23,24,25].

Host defense peptides are vital to kill a wide range of microorganisms in the defense barrier of innate immunity, such as β-defensin and cathelicidin in ruminants, responding to invading pathogens [26]. TAP is a β-defensin produced by bovine mucosal epithelial cells [27]. Bac5 is a proline- and arginine-rich cathelicidin that is stored as an inactive precursor (proBac5) in the large granules of bovine neutrophils [28]. In our work, the TAP and Bac5 mRNA levels in MAC-T cells were both increased after SPB stimulation. Interestingly, this upregulation was further enhanced when MAC-T cells were incubated with SPB and subsequently stimulated by LTA. By this inference, HDPs probably attribute the capacity to inhibit bacterial infection-induced inflammatory responses due to their strong affinity for bacterial membrane components such as lipoteichoic acid. Meanwhile, the reduction of *S. aureus* internalization into cells induced by SPB is likely due to HDP induction rather than the direct antibacterial or bacterial invasion-inhibitory activity of SCFAs [15,16]. For those bovine HDP genes that are induced by SPB, we also observed a clear pattern of gene-specific regulation, as evidenced by marked differences in the magnitude of induction among different HDP types. SPB increased mRNA levels by an average (mean) of 5–7 fold of TAP, but had a stronger effect on Bac5 (8–12 fold) with or without LTA treatment.

TLR2 is the major receptor of LTA. Activation of TLR2 leads to activation of NF-κB, which regulates the expression of inflammatory cytokines [29]. In this study, our results showed that SPB significantly inhibited *S. aureus* LTA-induced TLR2 expression, suggesting that SPB inhibited *S. aureus* LTA-induced inflammation by suppressing the TLR2 signaling pathway. The transcription factor NF-κB is a key regulator of inflammation and immune responses [30]. In normal cells, NF-κB is localized to the cytoplasm and bound to an inhibitory protein called IκBα. Once cells are treated with various inducers, IκBα is degraded, and phosphorylated NF-κB p65 is transferred from the cytoplasm to the nucleus and promotes the transcription of inflammatory cytokines [28]. Kim et al. [31] found that treatment with SPB substantially attenuated the pathophysiological features of LPS-induced lung inflammation, including inflammatory cell recruitment and vascular leakage, and reduced the nuclear translocation of NF-κB in lung tissue. A previous study showed that *S. aureus* internalization was associated with the active state of NF-κB, and inhibition of NF-κB activation could attenuate the internalization of *S. aureus* into bMECs [32]. In the present study, *S. aureus* LTA caused phosphorylation of NF-κB p65 and IκBα in MAC-T cells. Subsequently, we found that SPB could inhibit LTA-induced NF-κB activation in a dose-dependent manner. These results suggest that SPB inhibits NF-κB activation and reduces bacterial toxin-induced inflammation responses.

Activation of the NLRP3 inflammasome is observed in many inflammatory diseases, resulting in upregulation of the transcription of caspase-1 and proinflammatory cytokine genes [33]. During the infection of human macrophages by *Mycobacterium tuberculosis*, SPB independently regulates C-C motif chemokine ligand (CCL) secretion and multiple genes within the NLRP3 inflammasome pathway [34]. Kim et al. [31] also found that the expression of NLRP3, caspase-1, and IL-1β was substantially increased in lung tissue of LPS-instilled mice, and these increases were dramatically inhibited by administration of SPB. In this study, Western blot analysis revealed that the expression of NLRP3, ASC, caspase-1, and IL-1β proteins was increased during *S. aureus* LTA stimulation, while this increase was attenuated by pre-incubation with SPB in a dose-dependent manner. ASC, a mitochondrial antiviral signaling protein located in the outer membrane, may serve as a new adaptor to promote mitochondrial localization and activation of NLRP3 [33]. The NLRP3 inflammasome could promote the recruitment of ASC and procaspase-1, and result in the development of mature IL-1β [33]. As we assumed, SPB significantly suppressed *S. aureus* LTA-induced inflammation via attenuating ASC-dependent NLRP3 inflammasome activation.

## 4. Materials and Methods

### 4.1. Reagents and Antibodies

*S. aureus* LTA and SPB were supplied by Sigma-Aldrich (St Louis, MO, USA). All antibodies used in Western blots were purchased from Beyotime Institute of Biotechnology (Haimen, China). All other chemicals were reagent grade.

### 4.2. Cell Culture and Treatment

Bovine mammary alveolar (MAC-T) cells were a gift from Bu Dengpan (Chinese Academy of Agricultural Sciences, Beijing, China). Briefly, the MAC-T cell line was developed by immortalizing primary bovine mammary alveolar cells by stable transfection with a replication defective retrovirus (SV40) large T-antigen. MAC-T cells were cultured in basic Dulbecco’s modified Eagle’s medium supplemented with 5% heat-inactivated fetal bovine serum (FBS) and 1% penicillin/streptomycin (100 U/mL and 100 mg/mL, respectively) and maintained in a humidified atmosphere at 37 °C containing 5% CO_2_. Cells were routinely passaged at a rate of 80–90% for all experiments. Cells were pretreated with different concentrations of SPB (0.5, 2, 4 mM) for 24 h followed by incubation with 1 mg/mL LTA for 3 h. Phosphate-buffered saline (PBS)-treated cells served as a control.

### 4.3. Cell Viability Assay

The toxic effects of SPB on MAC-T cells were determined using the MTT assay. The cells were stimulated with SPB (0.5, 2, 4 mM) and LTA (1 mg/mL) for 24 h in a 96-well plate, followed by treatment with 20 μL of 5 mg/mL MTT for 4 h. Finally, the supernatant was removed and dissolved in 150 μL of DMSO in each well. The optical density was measured at 490 nm on a microplate reader.

### 4.4. Quantitative Real-Time PCR Analysis of Bovine Gene Expression

Total RNA was extracted from the cells using Trizol regent under all tested conditions and then used to synthesize cDNA. Quantitative real-time PCR (qPCR) was performed using a SYBR Premix Ex Taq™ kit (TaKaRa, Shiga, Japan) on a CFX96TM Real-Time PCR Detection System (Bio-Rad, Hercules, CA, USA). Gene expression levels were determined using the 2^−ΔΔCt^ method with bovine 18S rRNA as a reference. Specific primers are listed in Table 1.

### 4.5. Western Blot Analysis

Whole-cell lysates were extracted using RIPA Cell Lysis Buffer (Solarbio, Beijing, China) containing a protease inhibitor mixture (Solarbio, China). The concentration of protein in the extract from MAC-T cells was determined using a BCA Protein Assay Kit (Solarbio, China). Extracts containing equal quantities of proteins (50 μg) were resolved on 12% SDS-polyacrylamide gel and transferred to PVDF membrane and blocked with 5% non-fat dry milk in TBST for 2 h. The membrane was then incubated with primary antibody overnight at 4 °C. Primary antibodies were purchased from Beyotime Institute of Biotechnology (Haimen, China) at a 1:1000 dilution. After washing with TBST, the members were incubated with HRP-conjugated secondary antibodies at room temperature for 1 h. Finally, specific bands were produced using ECL reagent (Solarbio, China) and digitally detected with the GelDocTM XR Plus system (Bio-Rad, USA).

### 4.6. Statistical Analysis

All data were obtained from three independent experiments, performed in triplicate, and compared by analysis of variance (ANOVA). Results are shown as the means ± standard error (SE), and the level of statistical significance was defined as *p* < 0.05 or *p* < 0.01.

## 5. Conclusions

In summary, our study indicated that SPB significantly inhibited inflammatory cytokines and enhanced HDP expression in *S. aureus* LTA-stimulated MAC-T cells. The anti-inflammatory mechanism of SPB was through suppressing the TLR2, NF-κB, and NLRP3 signaling pathway induced by *S. aureus* LTA (Figure 6). These results indicate that SPB could be a potential agent for the treatment of mastitis.

## Figures and Tables

**Figure 1 molecules-23-03056-f001:**
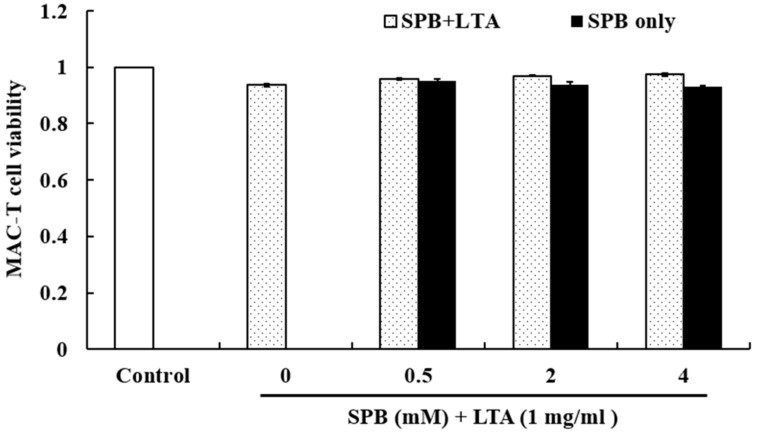
The effect of SPB on the cell viability of MAC-T cells. MAC-T cells were cultured in the presence of SPB (0.5, 1, 2, and 4 mM) and LTA (1 mg/mL) for 24 h, and then viability was determined by MTT assay. Each data point shows the mean ± standard error (SE) of triplicates from three independent experiments.

**Figure 2 molecules-23-03056-f002:**
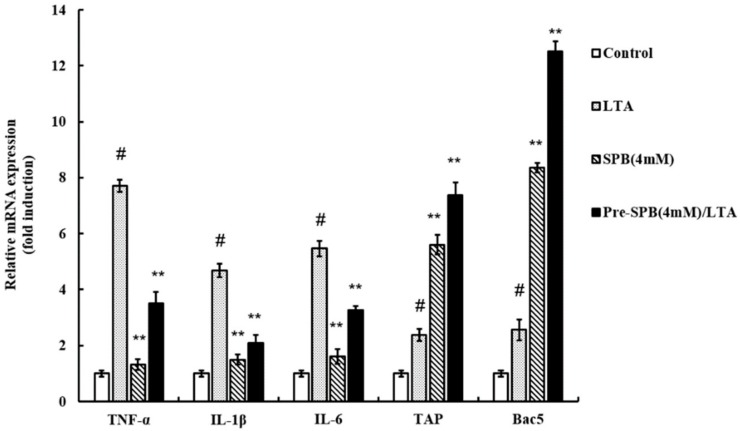
The effect of SPB on LTA-induced TNF-α, IL-1β, IL-6, TAP, and Bac5 production. Gene expression was normalized to the expression of the reference gene 18S rRNA. Each column shows the means ± SE of triplicates of three independent experiments. ^#^
*p* < 0.05 vs. control group; ** *p* < 0.01 vs. LTA group.

**Figure 3 molecules-23-03056-f003:**
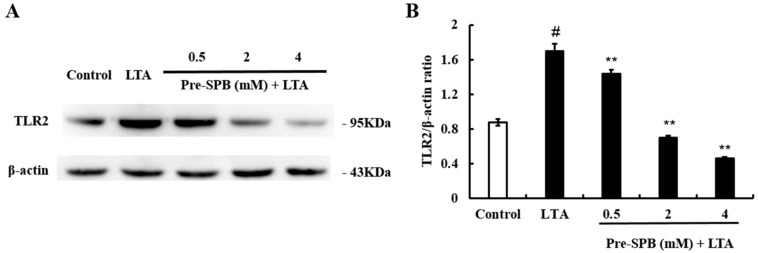
The effect of SPB on TLR2 expression. The expression of TLR2 was investigated in MAC-T cells using Western blot analysis. β-actin was used as a reference control. (**A**) The expression levels of toll-like receptor 2 (TLR2); and (**B**) The quantification histogram of TLR2 protein expression normalized by β-actin. The values presented are the mean ± SE of three independent experiments. # *p* < 0.05 vs. control group; ** *p* < 0.01 vs. LTA group.

**Figure 4 molecules-23-03056-f004:**
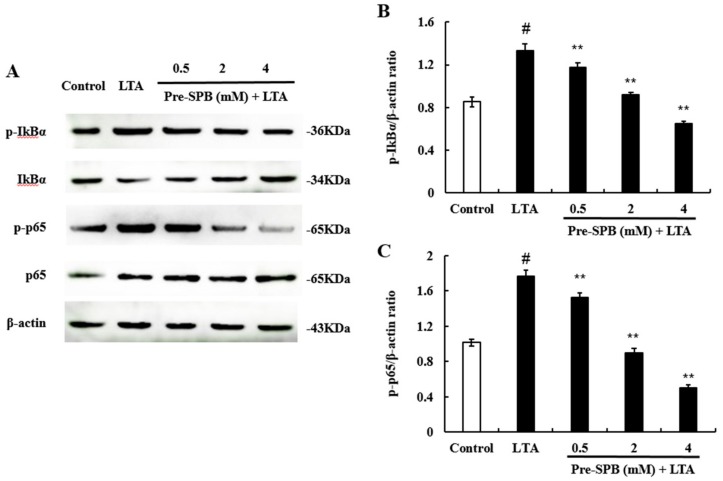
The effect of SPB on NF-Κb activation. The NF-κB pathway was explored in MAC-T cells using Western blot analysis. β-actin was used as a reference control. (**A**) The expression levels of p-IκBα, IκBα, p-p65 and p65; (**B**) The quantification histogram of p-IκBα protein expression normalized by β-actin; and (**C**) The quantification histogram of p-p65 protein expression normalized by β-actin. The values presented are the means ± SE of three independent experiments. # *p* < 0.05 vs. control group; ** *p* < 0.01 vs. LTA group.

**Figure 5 molecules-23-03056-f005:**
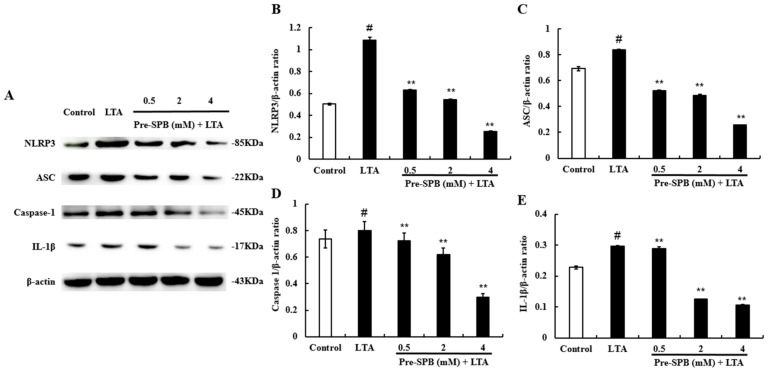
The effect of SPB on NLRP3 inflammasome. The NLRP3 inflammasome pathway was explored in MAC-T cells using Western blot analysis. β-actin was used as a reference control. (**A**) The expression levels of NOD-like receptor protein 3 (NLRP3), apoptosis-associated speck-like protein containing CARD (ASC), caspase-1, and interleukin 1 beta (IL-1β) proteins; (**B**) The quantification histogram of NLRP3 protein expression normalized by β-actin; (**C**) The quantification histogram of ASC protein expression normalized by β-actin; (**D**) The quantification histogram of caspase-1 protein expression normalized by β-actin; and (**E**) The quantification histogram of IL-1β protein expression normalized by β-actin. The values presented are the means ± SE of three independent experiments. # *p* < 0.05 vs. control group; ** *p* < 0.01 vs. LTA group.

**Figure 6 molecules-23-03056-f006:**
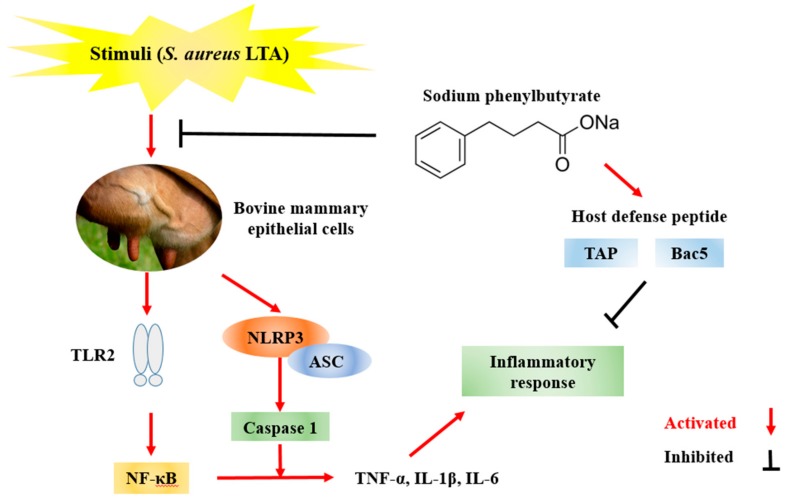
Schematic representation of protective effects of SPB against *Staphylococcus aureus* LTA-induced cell inflammatory responses in MAC-T cells.

**Table 1 molecules-23-03056-t001:** Primers used in this study.

Gene	Primer	Sequence (5′–3′)	Product Size (bp)
TNF-α	Forward	TGGGCTCCAAGCATCCAACT	182
	Reverse	GGCCCTCACATTCTGGGTGT	
IL-1β	Forward	TCGAAACGTCCTCCGACGAG	131
	Reverse	TGAGAGGAGGTGGAGAGCCT	
IL-6	Forward	CAAGCGCCTTCACTCCATTC	176
	Reverse	GATTTTGTCGACCATGCGCT	
TAP	Forward	CGCTCCTCTTCCTGGTCCTG	197
	Reverse	TGATCCCGGCTGTGTCTTGG	
Bac5	Forward	CAGTCACCCTGGACCCATCA	124
	Reverse	GGGCGGAACGGTGGATAGAA	
18S rRNA	Forward	AGTGGAGCCTGCGGCTTAAT	105
	Reverse	CACCACCCACGGAATCGAGA	

TNF-α—tumor necrosis factor alpha; IL—interleukin; TAP—tracheal antimicrobial peptide; Bac5—bactenecin 5; rRNA—ribosomal RNA.

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
