# Peer review of "Sodium Phenylbutyrate Ameliorates Inflammatory Response Induced by Staphylococcus aureus Lipoteichoic Acid via Suppressing TLR2/NF-κB/NLRP3 Pathways in MAC-T Cells"

_molecules, 2018, doi:10.3390/molecules23123056_

Reviewer 1 Report

I have reviewed an article: "Sodium phenylbutyrate ameliorates inflammatory 3 response induced by Staphylococcus aureus 4 lipoteichoic acid via suppressing TLR2/NF5 κB/NLRP3 pathways in MAC-T cells". Authors use lot of abbreviations, and they have listed all abbreviations at the end of the text, but specially in the Abstract and Introduction section it would be better to use a full name to make reading more understandable. Please, in Abstract section define what are MAC-T cells.

MM section, paragraph 2.2.: please, indicate from where did you obtain MAC-T cells.

Please, explain what was the rationale to test concentrations of SPB from 0.5-4 mM? Later, in Discussion section you have written that these concentrations did not show any toxic effect, but it seems to me that these concentrations are quite high.The same goes for LTA. Why did you choose to test 1 mg/mL of this compound? What is the connection between tested concentrations of SPB and LTA and expected concentrations of these compounds that can be found in realistic system?

2.3. paragraph: please indicate that you have tested cytotoxic effects of both, LTA and SPB.

Results section: please, on graphs indicate that concentration of LTA was 1 mg/mL.

Discussion section: please indicate what concentrations of LTA activated inflammatory reactions in cited paper (19) and compare (and comment) it with the concentration of LTA that was tested in this paper.

Line 178-please write the full name of HDP. 

Author Response

1. Comment and Suggestion: I have reviewed an article: "Sodium phenylbutyrate ameliorates inflammatory 3 response induced by Staphylococcus aureus 4 lipoteichoic acid via suppressing TLR2/NF5 κB/NLRP3 pathways in MAC-T cells". Authors use lot of abbreviations, and they have listed all abbreviations at the end of the text, but specially in the Abstract and Introduction section it would be better to use a full name to make reading more understandable. Please, in Abstract section define what are MAC-T cells.

Response: bovine mammary alveolar (MAC-T) cells

2. Comment and Suggestion: MM section, paragraph 2.2.: please, indicate from where did you obtain MAC-T cells.

Response: Bovine mammary alveolar (MAC-T) cells were a gift from Bu Dengpan (Chinese Academy of Agricultural Sciences, Beijing, China). Briefly, the MAC-T cell line was developed by immortalizing primary bovine mammary alveolar cells by stable transfection with a replication defective retrovirus (SV40) large T-antigen.

3. Comment and Suggestion: Please, explain what was the rationale to test concentrations of SPB from 0.5-4 mM? Later, in Discussion section you have written that these concentrations did not show any toxic effect, but it seems to me that these concentrations are quite high. The same goes for LTA. Why did you choose to test 1 mg/mL of this compound? What is the connection between tested concentrations of SPB and LTA and expected concentrations of these compounds that can be found in realistic system?

Response: A review of the literature related to butyrate infusion in ruminants identified an experimental treatment of 2 g /kg of BW of SPB as the greatest butyrate treatment administered to a ruminant (Neogrády et al., 1989). Both studies by Zarrin et al., (2013, 2014) showed plasma β-hydroxybutyrate averaged close to 3.0 mM and peaked close to 7 mM for cows administered the 2 g/kg of BW of SPB treatment. Herrick et al. (2016) found that the magnitude of change in plasma β-hydroxybutyrate between cows administered the control and 1 g/kg of BW of SPB treatment was 1.66 mM, and the change between the control and 2 g/kg of BW of SPB treatment cows was 2.47 mM. Based on references and MTT assay in our study, SPB (0.5, 2, 4 mM) has no cytotoxic effects on MAC-T.

Bougarn et al. (2010) investigated the capacity of bMECs to produce pro-inflammatory cytokines in response to LTA. Upon stimulation with S. aureus LTA at suboptimal concentrations (1 μg/ml), bMECs increased the level of mRNA expression of TNF-α, IL-1β except IL-6 at 2 h. Wellnitz et al. (2016) found that a cytotoxic effect (i.e., a considerably increased release of LDH into cell culture supernatant) was observed in bMECs challenged with 0.5 mg/mL LPS but not cells challenged with 20 mg/mL S. aureus LTA, even though this LTA treatment decreased the epithelial barrier integrity. Wellnitz et al. (2016) also found that bMECs challenged with 2 mg/mL LTA did not increase a cytotoxic effect, and did not decrease the mammary epithelial barrier integrity. Based on references and preliminary experiments, MAC-T cells significantly increased the level of mRNA expression of TNF-α, IL-1β and IL-6 at 3 h with S. aureus LTA stimulation at optimal concentrations (1 μg/ml) in our study.

Neogrády, Z., P. Gálfi, and F. Kutas. Effect of intraruminal butyrate infusion on the plasma insulin level in sheep. Acta Vet. Hung. 1989, 37, 247–253.

Zarrin, M., L. De Matteis, M. C. M. B. Vernay, O. Wellnitz, H. A.van Dorland, and R. M. Bruckmaier.Long-term elevation of β-hydroxybutyrate in dairy cows through infusion: Effects on feed intake, milk production, and metabolism. J. Dairy Sci. 2013, 96, 2960–2972.
Zarrin, M., O. Wellnitz, H. A. van Dorland, J. J. Gross, and R. M. Bruckmaier. Hyperketonemia during lipopolysaccharide-induced mastitis affects systemic and local intramammary metabolism in dairy cows. J. Dairy Sci. 2014, 97, 3531–3541.

Herrick, K.J., Hippen, A.R., Kalscheur, K.F., Schingoethe, D.J., Casper, D.P., Moreland, S.C., van, Eys J.E. Single-dose infusion of sodium butyrate, but not lactose, increases plasma β-hydroxybutyrate and insulin in lactating dairy cows. J Dairy Sci. 2017, 100, 757-768.

Bougarn, S., Cunha, P., Harmache, A., Fromageau, A., Gilbert, F. B., Rainard, P. Muramyl dipeptide synergizes with Staphylococcus aureus lipoteichoic acid to recruit neutrophils in the mammary gland and to stimulate mammary epithelial cells. Clin. Vaccine Immunol. 2010, 17, 1797-1809.

Wellnitz, O., Zbinden, C., Huang, X., Bruckmaier, R.M. Short communication: Differential loss of bovine mammary epithelial barrier integrity in response to lipopolysaccharide and lipoteichoic acid. J Dairy Sci. 2016, 99,4851-4856.

4. Comment and Suggestion: 2.3. paragraph: please indicate that you have tested cytotoxic effects of both, LTA and SPB.

Response: Considering the Reviewer’s suggestion, we have already maked some changes and marked in red in revised paper.

5. Comment and Suggestion: Results section: please, on graphs indicate that concentration of LTA was 1 mg/mL.

Response: Considering the Reviewer’s suggestion, we have already maked some changes and marked in red in revised paper.

6. Comment and Suggestion: Discussion section: please indicate what concentrations of LTA activated inflammatory reactions in cited paper (19) and compare (and comment) it with the concentration of LTA that was tested in this paper.

Response: Bougarn et al. (2010) investigated the capacity of bMECs to produce pro-inflammatory cytokines in response to LTA. Upon stimulation with S. aureus LTA at suboptimal concentrations (1 μg/ml), bMECs increased the level of mRNA expression of TNF-α, IL-1β except IL-6 at 2 h. Wellnitz et al. (2016) found that a cytotoxic effect (i.e., a considerably increased release of LDH into cell culture supernatant) was observed in bMECs challenged with 0.5 mg/mL LPS but not cells challenged with 20 mg/mL S. aureus LTA, even though this LTA treatment decreased the epithelial barrier integrity. Wellnitz et al. (2016) also found that bMECs challenged with 2 mg/mL LTA did not increase a cytotoxic effect, and did not decrease the mammary epithelial barrier integrity. Based on references and preliminary experiments, MAC-T cells significantly increased the level of mRNA expression of TNF-α, IL-1β and IL-6 at 3 h with S. aureus LTA stimulation at optimal concentrations (1 μg/ml) in our study. 

Bougarn, S., Cunha, P., Harmache, A., Fromageau, A., Gilbert, F. B., Rainard, P. Muramyl dipeptide synergizes with Staphylococcus aureus lipoteichoic acid to recruit neutrophils in the mammary gland and to stimulate mammary epithelial cells. Clin. Vaccine Immunol. 2010, 17, 1797-1809.

Wellnitz, O., Zbinden, C., Huang, X., Bruckmaier, R.M. Short communication: Differential loss of bovine mammary epithelial barrier integrity in response to lipopolysaccharide and lipoteichoic acid. J Dairy Sci. 2016, 99,4851-4856.

7. Comment and Suggestion: Line 178-please write the full name of HDP.

Response: Host defense peptides (HDP)

Reviewer 2 Report

Studies of Zhang and co-authors showed that SPB could be a potential agent for the treatment of mastitis. The results of the research are valuable and interesting. The manuscript should be published. However, in Introduction section information about Staphylococcus aureus bacteria should be extended

Author Response

Comment and Suggestion: Studies of Zhang and co-authors showed that SPB could be a potential agent for the treatment of mastitis. The results of the research are valuable and interesting. The manuscript should be published. However, in Introduction section information about Staphylococcus aureus bacteria should be extended.

Response: Considering the Reviewer’s suggestion, we have added some information about Staphylococcus aureus bacteria in introduction section and marked in red in revised paper as follows:

It colonizes the mammary gland tissues by internalizing epithelial and endothelial cells. S. aureus produces virulence factors such as enterotoxins, toxic shock syndrome toxin, and panton-valentine leucocidin, which can stimulate severe host cell inflammatory responses and enhance the expression of various pro-inflammatory mediators in bovine mammary gland epithelial cells [3]. Although the expression of pro-inflammatory cytokines is important for eliminating an invasive bacteria, excessive production of these pro-inflammatory mediators often leads to a continuous inflammatory response in bovine mammary glands, possibly resulting in chronic mastitis [4].

[3] Bonsaglia, E.C.R.; Silva, N.C.C.; Rossi, B.F.; Camargo, C.H.; Dantas, S.T.A.; Langoni, H.; Guimarães, F.F.; Lima, F.S.; Fitzgerald, J.R.; Fernandes, A. Júnior; Rall, V.L.M. Molecular epidemiology of methicillin-susceptible Staphylococcus aureus (MSSA) isolated from milk of cows with subclinical mastitis. Microb. Pathog. 2018, 124, 130-135.

[4] Sadek, K.; Saleh, E.; Ayoub, M. Selective, reliable blood and milk bio-markers for diagnosing clinical and subclinical bovine mastitis. Trop Anim Health Prod. 2017, 49, 431-437.